# Carers' and health workers' perspectives on malnutrition in infants aged under six months in rural Ethiopia: A qualitative study

Nega Jibat[1]*, Ritu Rana[2,3], Ayenew Negesse[4], Mubarek Abera[5], Alemseged Abdissa[6,7], Tsinuel Girma[8,9], Anley Haile[10], Hatty Barthorp[2], Marie McGrath[11], Carlos S. Grijalva-Eternod[12,13], Marko Kerac[12], Melkamu Berhane[8]

1 Department of Sociology, Jimma University, Jimma, Ethiopia, 2 GOAL Global, Dublin, Ireland, 3 Indian Institute of Public Health, Gandhinagar, India, 4 Department of Human Nutrition, Debre Markos University, Debre Markos, Ethiopia, 5 Department of Psychiatry, Jimma University, Jimma, Ethiopia, 6 Department of Laboratory and Microbiology, Jimma University, Jimma, Ethiopia, 7 Armaeur Hansen Research Institute (AHRI), Addis Ababa, Ethiopia, 8 Department of Paediatrics and Child Health, Jimma University, Jimma, Ethiopia, 9 Harvard Chan School of Public Health, Addis Ababa, Ethiopia, 10 GOAL Ethiopia, Addis Ababa, Ethiopia, 11 Emergency Nutrition Network, Oxford, United Kingdom, 12 London School of Hygiene and Tropical Medicine, London, United Kingdom, 13 UCL Institute for Global Health, London, United Kingdom

* negajibat@gmail.com

**Data Availability Statement:** All relevant data are within the manuscript and its Supporting Information files.

## Abstract

### Objectives

Supporting small and nutritionally at-risk (potentially malnourished) infants under six months is a global health priority, albeit with a weak evidence-base. To inform policy and research in this area, we aimed to assess the perceptions and understanding of infant malnutrition and its management among carers, communities, and healthcare workers in rural Ethiopia.

### Methods

We conducted in-depth and key-informant interviews, from May-August 2020 in Jimma Zone and Deder District, Ethiopia. We used purposive sampling to recruit the participants. Interviews were transcribed into Amharic or Afaan Oromo and then translated into English. Atlas ti-7 was used to support data analysis. Findings were narrated based on the different themes arising from the interviews.

### Results

Carers/community members and healthcare workers reported on five different themes: 1) Perceptions about health and well-being: an 'ideal infant' slept well, fed well, was active and looked 'fat'; 2)Perceptions of feeding: overall knowledge of key recommendations like exclusive breastfeeding was good but practices were suboptimal, notably a cultural practice to give water to young infants; 3)Awareness about malnutrition: a key limitation was knowledge of exactly how to identify small and nutritionally at-risk infants; 4) Reasons for malnutrition: levels of understanding varied and included feeding problems and caregiver's work pressures resulting in the premature introduction of complementary feeds; 5) Perceptions about

**Funding:** This grant is held by LSHTM (PI MK) Grant reference: EPPHZR37 Full name of funder: The Eleanor Crook Foundation http://eleanorcrookfoundation.org/ The funder had no role in study design, data collection and analysis, decision to publish, or preparation of the manuscript.

**Competing interests:** The authors have declared that no competing interests exist.

**Abbreviations:** COVID-19, Corona virus Disease; HEW, Health Extension Worker; HP, Health Post; MAMI, Management of Small and Nutritionally At-Risk Mothers and Infants U6m; MCH, Maternal and Child Health; MUAC, Mid-Upper Arm Circumference; u6m, Under Six Months; UNICEF, United Nations International Children's Emergency Fund; WHO, World Health Organisation; cRCT, Cluster Randomised Controlled Trial.

identification & treatment: carers prefer treatment close to home but were concerned about the quality of community-based services.

## Conclusion

To succeed, research projects that investigate programes that manage small and nutritionally at-risk infants under six months should understand and be responsive to the culture and context in which they operate. They should build on community strengths and tackle misunderstandings and barriers. Interventions beyond just focusing on knowledge and attitude of the carers and health workers are necessary to tackle the challenges around infants under 6 months of age at risk of malnutrition. Moreover, stakeholders beyond the health sector should also be involved in order to support the infants under 6 months and their mothers as some of the key reasons behind the at-risk infants are just beyond the capacity of the health sector or health system. Our list of themes could be used to inform infant nutrition work not just in Ethiopia but also in many others.

## Introduction

Improving the management of "small and nutritionally at-risk" infants aged under six months (henceforth 'malnourished' infants u6m) is a global health priority [1, 2]. An estimated 8.5 million infants u6m worldwide are wasted (have low weight-for-length), of these, 3.8 million are severely wasted ─ a marker of severe malnutrition where outcomes are poorest) [3].Because of physiological and immunological immaturity, they are at high risk of disease and death in the short term [4, 5]. They are also at increased risk of non-communicable diseases in later life [6].

Compared to older children, supporting malnourished infants u6m is complex as there are many potential underlying problems, e.g. low birth weight, congenital disease, disability, infection, difficulties with exclusive breastfeeding [1]. Supporting breastfeeding is at the core of treatment for this age group. Still, support is often complex and time-consuming, requiring skilled professionals and a conducive environment, especially when a mother-infant dyad is experiencing difficulties [7]. Due to these challenges, adequate management of malnourished infants u6m often lags behind older children aged 6–59 months. For older malnourished children, community-focused care has long been standard [8, 9] whereas for infants, inpatient-based guidelines are still dominate [10], limiting nutrition programe cover age and limiting impact.

In 2013, updated World Health Organization (WHO) guidelines for "The Management of Severe Acute Malnutrition in Infants and Children" included, for the first time, a stand-alone chapter on infants u6m [11]. This chapter had recommendations for community-based care for small but clinically stable infants u6m (uncomplicated malnutrition). However, the guidelines' recommendations for infants were based on low-quality evidence, a significant limitation, as research in this area remains sparse.

Ethiopia is among the countries where infant u6m nutrition status is a concern. In the 2019 Ethiopian Mini Demographic and Health Survey,17.1% of infants u6m were stunted, 9.8% underweight, and 9.6% wasted [12].

While exclusive breastfeeding was practiced by 59% of mothers overall, declining sharply with age: 73% in 0–1 month to 40% in 4–5 months [12]. Contrary to WHO recommendations on exclusive breastfeeding for u6m, 14% of infants were given plain water, 13% were given

complementary foods, 8% were given other milk, 6% were never breastfed; 9% were given a bottle with a nipple. As in many r low- and middle-income countries, the Ethiopian Ministry of Health national guideline for managing severe acute malnutrition still recommends inpatient-only treatment for infants u6m [13].

To address the lack of evidence both in Ethiopia and internationally, we are planning to conduct a cluster randomised controlled trial (cRCT) to assess the effectiveness of a new Management of small and nutritionally At-risk Infants u6m and their Mothers (MAMI) Care Pathway Package [14]. This uses an integrated care pathway approach for adaptation and integration across existing systems and services [15].To optimise impact, the MAMI Care Pathway needs to be socially and culturally acceptable and it is vital to understand the current situation around infant u6m nutrition. To inform both our study and work in other settings, and support intervention programs for infants u6m at risk of or with malnutrition, understanding the current situation around infant u6m is essential; the aim of this qualitative study is thus to assess the perceptions and understanding of malnutrition in infants u6m and its management among carers, communities and healthcare workers in rural Ethiopia.

## Methods

### Study setting

We undertook this study in Jimma Zone and Deder District of Eastern Hararghe Zone (where our future cRCT is being planned). We conducted fieldwork from May to August 2020. The sites were chosen to represent geographic areas considered relatively food secure (Jimma) and emergency/food insecure (Deder). Jimma Zone, located in south- western Ethiopia, is one of the populous zones in Oromia National Regional State. While well known for its agricultural productivity (the primary product being coffee), it is also one of the areas of the country with a high burden of malnutrition. Deder District is located in Oromia National Regional State, East Hararghe Zone. Main livelihoods in the area are agriculture, petty trade of cash crops such as khat and coffee, fattening of oxen and local casual labour.

### Domain 1- Research team and reflexivity

**Personal characteristics.**   Our research team includes authors from diverse professional, socio-economic and cultural back grounds from Ethiopia, the United Kingdom (UK) and India. This includes paediatricians, nutritionists, public health experts, a microbiologist and a sociologist; male and female; academics and practitioners; a range of seniority from MSc/MA to Professors. All have prior experience working on related health issues in Ethiopia or other countries with similar contexts. Interaction among the researchers from different countries with different research traditions created a learning and experience- sharing platform.

**Relationship with participants.**   Ethiopian team members who work in and directly understand the local context led the data collection and first-line analysis of study results. Non-native team members had exposure to the study communities through field visits, stakeholder consultation and previous projects and contributed to results interpretation. Some of the data collectors belong to the dominant ethnic group of the study sites ─ the Oromo, others do not, however, none of them is local member of the target communities. The data collectors are fluent in Afaan Oromo or Amharic, with different religious affiliations. One of the data collectors is a clinician (paediatrician) but he did not interview his patients. Others do not have direct service relations with the study participants.

## Domain 2- Study design

**Theoretical framework.** This was an in-depth and key-informant interview-based study using a thematic analysis framework.

**Participant selection.** Our target participants, recruited using purposive samples via heads of health facilities, health workers and health extension workers (HEWs) were:

- (1) **Carers and community members**: mainly mothers, fathers and health development army (HDA) members. Mothers with infants u6m with malnutrition (3) and mothers of healthy infants u6m (7) were included in the interview. Fathers and mothers were selected from different households. HDA is an organized movement of communities forged through participatory learning and action meetings scaling up best practices. All the HDAs are females.

- (2) **Healthcare workers:** nurses, midwives, health officers, paediatricians, and HEWs recruited from local health facilities (health posts, health centers and hospitals). These healthcare workers managed small and sick infants were recruited from 7 health facilities. All of the HEWs are females while the others are mixture of females and males.

- (3) **Health managers and program officers:** included personnel at the district and zonal level health departments are heads of the family health units at their respective offices. They work on different programs including infant and young child feeding (IYCF) and infant and child malnutrition. They also support the health facilities and health care workers at these facilities towards accomplishing the objectives of different programs. These cadres were recruited from 4 offices (2 district health offices and 2 zonal health offices).

All study participants were selected based on their experiences and knowledge pertinent to the study problem, and their willingness to participate. All participated in a face-to-face key-informant interview or in-depth interviews that were conducted face to face. Data collection was carried out in private at home and workplaces (offices and health facilities). The sample size was determined by data saturation (i.e., when no new topics were emerged from interviews). Previous studies on this theme in other settings achieved data saturation after approximately ten interviewees per group. The time schedule was arranged based on the participant's convenience.

**Data collection.** A team consisting of an experienced sociologist, paediatrician and child nutrition expert was involved in data collection. The paediatrician did interview carers of sick infants in Jimma Medical Center where he provides clinical services to avoid influence of the patient-physician relationship during the interview and affect data quality. The child nutrition expert was involved in interviews with Amharic-speaking participants given he does not command Afaan Oromo. We conducted interviews using semi-structured interview guides translated into the two main local languages, Amharic and Afaan Oromo. We based our interview guide on previous work on small and nutritionally at-risk (malnourished) infants u6m in other settings [16] and piloted it prior to use in this setting. Due to participant time constraints, we did not repeat interviews and did not return transcripts to the interviewees for correction. We audio-recorded interviews on secure password-protected tablets (Samsung Galaxy Tab A) to enable later transcription. The lead interviewer also made field notes. The average duration of the interview was 45 minutes.

We followed all the COVID 19 preventive measures suggested by WHO and the Ethiopian Ministry of Health during data collection. To be specific, the data and the study participants have used medical/surgical masks all times during the interviews; physical distance of at least 1meter between the data collectors and the study participants were maintained and hand

hygiene using alcohol-based sanitizer was practiced whenever needed. As part of the study, we provided all the supplies (masks, sanitizers, etc.) to the study's data collectors and study participants.

### Domain3: Data analysis

We transcribed data to Afaan Oromo or Amharic and then translated it into English. We initiated data transcription during the data collection, but we transcribed the majority of the data soon after completing of the fieldwork. We coded and analyzed transcribed data thematically, emphasizing to any newly emerging themes. In cases of duplication, we merged duplicated codes or themes. We refined the codes and themes through several readings of the transcripts and further examination of the themes arising. A single coder per transcript initially coded the data that were shared with other team members for their reflections and feedback. The agreed-upon versions were analysed. Experienced qualitative researchers conducted data processing and analysis using ATLAS.ti software (version 7.5.18, Educational multiuser licence 25 units @1993–2021 byATLAS.ti, GmbH, Berlin). We presented the findings in narratives by thematic areas. The quotes included in the results were typical views expressed in each interview to exemplify emerging themes.

### Ethical considerations

The Jimma University's Institutional Review Board (Ref.No.IHRPGD/455/2020) and the London School of Hygiene and Tropical Medicine Ethics Committee (Ethics Ref: 17998) provided ethical clearance for the study. We obtained written consent from all study participants with the language of their interest.

## Results

### Socio-demographic characteristics of the study participants

We conducted 31 interviews, 17 with carers/community and 14 with healthcare workers and managers as key-informants. Mothers, fathers and HDA leaders represented carers and the community. Health workers including HEWs, nurses, health officers and paediatricians constituted the healthcare providers' side. A smaller number of participants from Deder than Jimma resulted from the proportional representation of the two study sites given only a district was considered in the former. Most participants we approached gave consent and agreed to be interviewed. Exceptions included a father who was caring for a malnourished infant in Jimma Medical Centre declined to be interviewed after showing initial interest. Lack of interest to give written consent was the reason for decline. Similarly, a female nurse from the same health facility refused to participate because the required time for the interview would affect her regular duties. In Table 1, we present the summary of the study participants by category and site.

The following themes emerged from the interviews and are summarised in Fig 1.

### Theme1: Perceptions about health and well-being of infants u6m: An "Ideal infant"

**Community & caregivers' perspectives.** Caregivers' perceptions of health and wellbeing focused on externally observable physical appearance, behaviours and lack of illness. They understood an infant as healthy and well if the infant appears to be "fat" which refers to big body size rather than more of fat tissue, is growing well, and behaviourally seems to be less disruptive, sleeps and feeds well, and has no diarrhoea or other symptoms of illness like fever and unable to suck. One study participant said:

**Table 1. Summary of study participants.**

| Participant Category | Total number of Interviews by category | Study site | |
|---|---|---|---|
| | | Jimma | Deder |
| ***Carers/Community members*** | **17** | **11** | **6** |
| Mothers | 10 | 7 | 3 |
| Fathers | 4 | 2 | 2 |
| Health Development Army | 3 | 2 | 1 |
| ***Healthcare workers (HCW):*** *Health extension workers, nurses, health officers, paediatricians* | **10** | **8** | **2** |
| ***Healthcare managers*** District Health Office and Zonal Health Office staff(Family health unit) | **4** | **2** | **2** |
| **TOTAL** | **31** | **21** | **10** |

*If an infant does not properly feed his mother's breast, persistently cries, and becomes restless, he is not feeling well;, or is not healthy. . .by looking at physical appearance, we can judge whether the infant is healthy. If the infant looks nice and smiles, we can say that he is healthy.* (Deder Father)

Sleep was highlighted as necessary, adequate sleep being frequent but not excessive:

*A healthy infant should sleep for 3–6 hours a day (*refers to 12 hours daytime*) at different intervals. It is not good if an infant remains asleep throughout the day. So, a very long hour of sleep is not good for infants.* (Jimma HAD Leader)

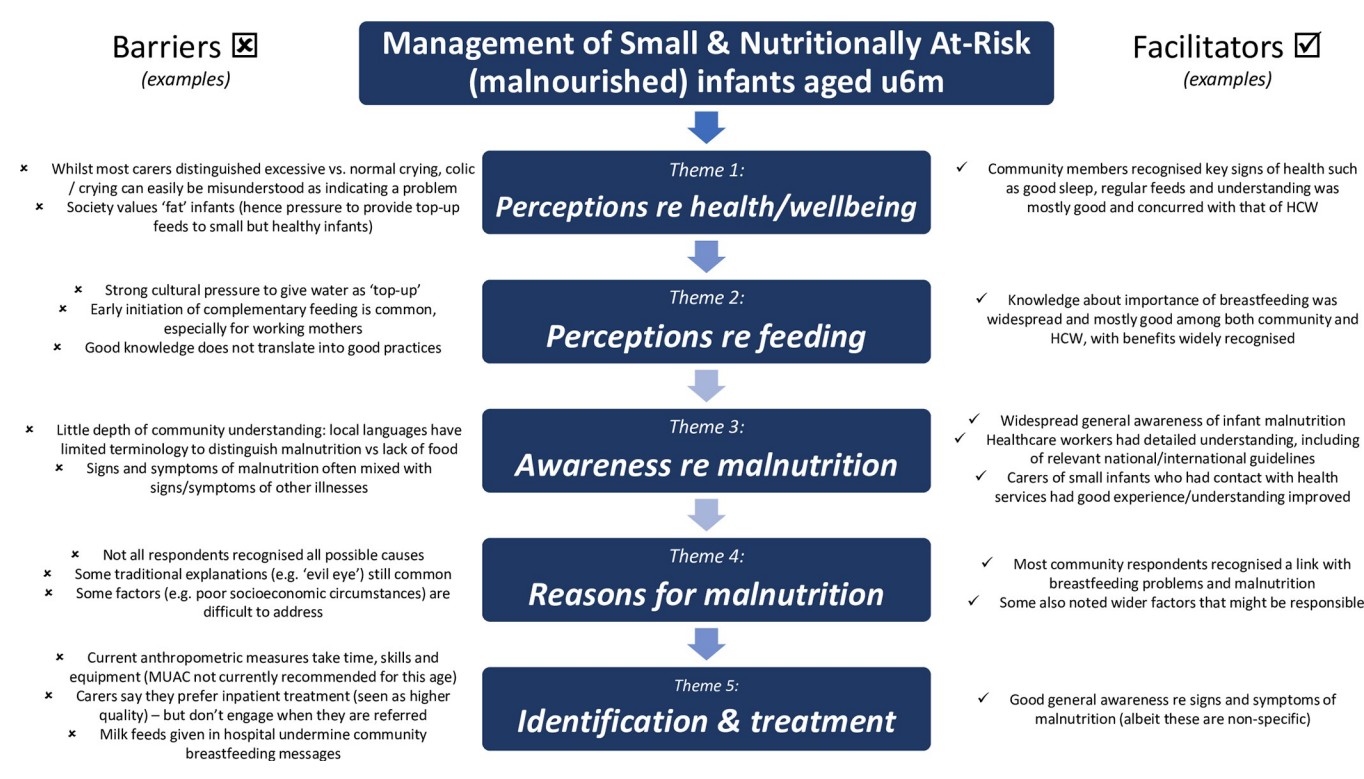

**Fig 1. Summary of infant malnutrition themes.**

*The infant should get adequate sleeping. In my opinion, a healthy infant should fall asleep for an hour at one time. But, if the infant remains asleep the whole day, it is not healthy sleeping.* (Deder Mother)

Being alert and active was also seen as a sign of good health. Feeding frequency was noted. Caregivers perceived that healthy infants feed well and frequently, with some2-5timesperday-being seen as adequate day time feeding frequency:

*The infant should feed his mother's breast about five times daily. An infant should feed his mother's breast until he is fully satisfied. (*Deder Mother)

Burping was described as a sign of wellbeing and satisfaction with the feeding. Community participants distinguished posseting (regurgitation of small amounts of milk) from vomiting, the former being normal and not a problem and the latter being a potential sign of illness:

*Posseting is also good for infants. It is a sign of satisfaction with the mother's breast milk. Itmay also indicate that the food is tolerated in the infant's digestive system* (Jimma HDA Leader)

*Burping is good as it indicates that the infant is satisfied with the feeding and the feeding is getting tolerated by their stomach, but vomiting is a sign of illness.* (Dede rMother)

Mild crying was considered normal but moderate to high intensity of crying was considered as a sign of pain, hunger, need for feeding or a sign of other problems: For example, a mother noted that:

*It is normal for infants to cry sometimes. But if he cries continuously, it may indicate that the infant is not feeling well, or he is feeling pain.* (Deder Mother)

Physical appearance and growth were also recognised as visible markers of health. An infant increasing in size (length and fatness) is considered to be healthy:

*We can see physical growth in infants if one gets bigger or gets fat and looks good. However, if the child is getting thin and bone, we say he is not showing the physical change.* (Jimma Father)

Community members also recognise *abnormal patterns of growth;* for example, an increase in abdominal size alone can be a sign of underlying illness and is abnormal:

*We can say that an infant is on the right track of physical growth if his whole-body parts are growing together. However, if his abdomen is bigger than other body parts, that is not a normal growth.* (Jimma HDA Leader)

Wider circumstances conducive to good infant health and wellbeing were also recognised, notably
Maternal conditions before and after delivery:

*A healthy infant is an infant who has received proper care starting from the day one of conception in the womb till its birth. As well, if the mother gets proper care and treatment during postnatal care, she can have a healthier infant. (HDA leader, Deder)*

**Healthcare workers' (HCWs) perspectives.** According to HCWs, healthy infants have balanced weight-for-age, sleep 2–3 times a day, smile, play, have good body movements, show happy faces and emotions, and usually grow. At the same time, persistent crying indicates a health problem. An ideal infant should breastfeed 8–12 times/24 hours until satisfied. As with caregivers, posseting was considered normal and a sign of good health and satisfaction with breastfeeding. Burping was noted as necessary to reduce reflux and aspiration after each feed.

*We say a child is healthy if her age and weight are congruent when weight to age measurement is in the normal range. U6m infant should get adequate sleeping. But if he is sleeping too much and not waking up for feeding, we would say that it is not normal sleeping.. . . A sick infant cries day and night. A normal child may cry when he gets hungry. Posseting is good for infants, indicating that the infant is satisfied. (Jimma HEW)*

Another HCW emphasised maternal conditions before and during pregnancy and after delivery: *For me, an ideal infant is the one born to a mother who had a maximal care during pregnancy, and adequate follow up during pregnancy and delivery; an infant who received all the necessary routine cares during early infancy, breast milk within the first one hour, exclusive breastfeeding with a frequency of 8–12 times, with proper positioning and attachment, and vaccination according to the schedule.* (Jimma Paediatrician)

HCWs also emphasised the importance of continuous growth as per the growth chart measurements and noted the achievement of developmental milestones in the first 1000 days of life. Finally, HCWs were keen to comment on what they saw as common misunderstandings and myths about infants' and mothers' health and wellbeing. e.g., in Deder, there is a community belief that eating animal products and fruits will lead to excessive foetal growth and hence predispose to caesarean section delivery.

## Theme2: Perceptions about the feeding of infants u6m

**Community & caregivers' views.** Awareness about appropriate feeding of infants u6m was reflected by the widespread use of the phrase "*Everybody knows.*" Nearly all study participants expressed a good understanding of the importance of exclusive breastfeeding up to six months, the reasons for this and the negative impact of not doing so:

*Everybody knows that u6m infants don't feed something else than breastfeeding.* (Father, Jimma)

*Breast- feeding is good for maintaining the health and wellbeing of an infant. Breast milk is complete as it contains water, food, medicine, and other necessary nutrients for the baby. So, breastfeeding is essential for u6m infants. Therefore, mothers should feed their u6minfantsonlywith breast milk.* (Deder Mother)

Many mothers told of exclusive breastfeeding being good for the physical strength and wellbeing of infants, due to mothers' breast milk "being complete." Some are also reported that breast milk contains medicine in addition to water and foods.

HDA leaders from both study sites also expressed the benefits of breastfeeding and revealed that key messages had come to them via community-based HEWs:

*Breast milk is enough for up to 6 months of age and it is not necessary to give something else. You just need to breastfeed infants 8–12 times per day properly. The HEWs taught us like that.* (Jimma HDA Leader)

*All community members have learned about the benefits of exclusive breastfeeding foru6minfants. Mothers also have good awareness about lactation techniques. Thus, all women in our community have good knowledge.* (Deder HDA Leader)

Mothers knew about official infant feeding recommendations and strongly valued them. They however noted some practical difficulties in following that advice. This was due to some discordance between some community customs/beliefs and the recommendations:

*People who resist exclusive breastfeeding until six months believe that mother's breast milk alone is not sufficient to maintain the health of u6m infants as a reason for early initiation of additional foods to u6m infant.* (Jimma HDA Leader)

While most were still reported as conforming to the recommendations, some adverse practices were common, e.g., early initiation of additional feeding by some caregivers remains a significant concern among healthcare providers.

**Reasons for early initiation of additional feedings.** Many factors were reported as driving factors of the early initiation of additional feedings:

- **Mothers' health and economic conditions:** were seen as the significant factors. Economic factors were especially pressing in Deder where mothers have to balance financial roles (e.g., selling khat in local markets) with their infants' caring roles. Given that men control farmland and its products (mainly khat in this context), women are forced to engage in trade to feed their families, which is considered a women's domain. As a result, mothers experience stressful life and physical exhaustion, significantly compromising maternal health and well-being. These pressures contribute to the initiation of additional feedings, to make up for lack of time to breastfeed.

   **'No life without water':** Although reportedly declining, many reported strong cultural pressures to introduce additional feeds before six months of age. Notably, it is believed that water prevents and heals constipation. Advice from grandmothers, mothers-in-law, and neighbours pressures mothers to introduce water. It is linked to beliefs that "humans cannot survive without water" and that "breast milk has a burning effect of causing thirst" and that "breast milk does not contain sufficient water." These beliefs persist regardless of HCW's education efforts.
   Unwillingness to give water to an infant 'in need' is regarded by many in the community as strange and indicative of lack of empathy for the infant and failure of meeting one's expected parental roles. Some even see giving water as a moral duty. Referring back to their own and the parents' past experiences social and cultural influencers blame 'modern' parents' refusal of giving water as an immoral act. The strength and deep-rootedness of beliefs around water (as well as the clash with modern teachings) are evidenced by reports of mothers with malnourished infants in health facilities giving water to infants–but hiding this from supervising HCW.
   Concerning to moral/religious accountability of parents to give water to their infants, a study participant had to say,

   *The people believe that the infants will accuse them to God -'Rabbitti nu himatti' and God will punish them for refusing infants to drink water—'Rabbi irraa nu gaafata'.* (Jimma HDA Leader)

- **Normalization of the feeding initiation as natural timing:** A common community belief that infants cry for food when others feed initiated by biological need and respond well to complementary feeding even if this happens before six months is normal. Even mothers who

believe in the importance of exclusive breastfeeding are told that waiting for the completion of six months to initiate additional feeding could have adverse effects on the future feeding patterns of the infant. The most typical concerns are future loss of appetite and refusal of supplementary feeding after six months. The latter is widespread fear among employed mothers. There is also a belief that infants who did not start additional feeding until six months remain thin as they grow in the future; hence, some parents deliberately start complementary feeds at five months of age. This fear of remaining thin is reinforced by society valuing fat infants as healthy and well. Beliefs about permanent loss of appetite and staying thin are expected, including among educated parents. They are also common among some healthcare providers. A mother's stand about the importance of initiating additional feeding after four months is,

*An infant should be exclusively breastfed for four months; it is acceptable. After that, I don't see its importance; it is just a lie. Health workers say 'the infant should exclusively breastfed for six months', but breast milk is not sufficient after four months as the need of the infant increases with age. I feel a health worker comes after the mother. It is the mother who knows better about her infant's needs.* (Jimma Mother)

**Healthcare workers' (HCWs) views.** HCWs views about u6m infants' feeding are clear and straightforward: exclusive breastfeeding must be practiced except in rare and special cases where this is not appropriate like when the mother is missing or unable to breastfeed because of her critical illnesses or when the infant cannot feed on breast milk because of congenital health problems. There was no or little difference in the extent of awareness among HCWs at different levels with regards to importance and techniques of exclusive breastfeeding, a paediatrician in Jimma summarizing typical views as:

*Except for special cases like mother's illness, infants u6m should exclusively feed on the mother's breast milk. But sometimes science recommends using formula milk, which is commercially introduced as food for infants u6m when a mother could not breastfeed her infant for different reasons. Many mothers are aware of exclusive breastfeeding. However, some mothers tend to give other foods to their infants when they reach four months. (*Jimma Paediatrician).

All HCWs had a clear understanding and firm stance that mothers should feed their infants u6m with only breast milk regardless of their views otherwise. All HCW strive to support breastfeeding through advice, counseling and educating caregivers. Their efforts, but mixed results are shared below:

*We teach mothers on the importance of breastfeeding for u6m infants. . . . We also teach mothers the appropriate lactation skills. As a result, these days, mothers are aware of the principles of exclusive breastfeeding. . .. However, some mothers do not apply this. Some mothers complain that their breast milk alone is insufficient to satisfy their infants. Such mothers might be tempted to give other foods like water mixed with cow milk.* (Deder HEW)

Given the consistency of views among HCWs regarding their knowledge and values about infant feeding, examining their views and observations about caregivers' situations is important. They agreed that mother's widespread knowledge about the advantages of exclusive breastfeeding did not translate into practice and highlighted the challenges of this discrepancy:

*In short, people's awareness is good but translating them into practice remains a big challenge. I can assure you that mothers believe 100% in the importance of breastfeeding. They know*

*pretty well about the benefits of exclusive breastfeeding for u6m infants. However, They don't live up to those convictions.* (Deder Health center nurse)

Among the factors contributing to the early initiation of complementary feeding is the short duration of maternity leave. This impacts even those mothers who would like to breast-feed for the entire six months exclusively:

*Practicing exclusive breastfeeding is still a problem for mothers who are government employ-ees. Now, the maternity leave is increased from 3 to 4 months. But still, it is not enough. Work-ing mothers can only stay at home with their babies for up to 4 months. So, the remaining two months for exclusive breast feeding up to six months is still a challenging issue. That is why many working mothers, including health professionals, start early to feed additional foods for their infants. . .. The concerned bodies should fill this gap between the principles of exclusive breastfeeding and maternity leave for working mothers.* (Jimma Health Officer)

## Theme3: Awareness about the occurrence of malnutrition among infants u6m

**Caregivers and community awareness.** Almost all caregivers were aware of malnutrition among infants u6m. However, there were varying levels of understanding. Two issues are nota-ble. First, interchangeably using the exact phrase roughly translated as "shortage of food" to denote the difficulty of getting "edible substance—food" and "nutritional status" in local lan-guages makes it challenging for people to understand the meaning of "malnutrition" differ-ently from starvation. Some caregivers have difficulty to imagining how their children could be malnourished so long as they feed them daily regardless of the type of food. Second, those who practice or believe in practicing exclusive breastfeeding tend to consider questions about malnutrition in infants u6m as irrelevant so long they are not eating food other than breast milk. Parents who had small or malnourished infants better understand related concepts because of their contact with health facilities during treatment.

*Yes, it occurs. For instance, this child was malnourished and was transfused with blood. One infant was faced with malnutrition in the community after the early initiation of complemen-tary feeding. (*Jimma mother)

*Yes, it affects, it has affected my infant as well. I gave him cow's milk at the age of 20 days while I shouldn't have given him. (*Jimma mother)

## Theme 4: Reasons about the occurrence of malnutrition among infants u6m

**Caregiver and community perceptions.** Caregivers had different levels of understanding. Most reported that inadequate and inappropriate breastfeeding (not feeding 8–12 times per day) as themaindriversofmalnutritionamonginfantsu6m.

*The root cause of malnutrition among u6m infants is insufficient breastfeeding. . . For u6m, we cannot talk about a balanced diet apart from breast milk. It is expected that the mother's breast milk contains all the necessary components of the diet. So, if the infants received appro-priate breastfeeding, malnutrition could not occur. (*Jimma mother)

Inadequate breast milk was said to occur due to insufficient and poor maternal diet quality. Other causes of malnutrition people mentioned include:

- Illnesses such as diarrhea and vomiting;

- Unhealthy baby at birth;

- Compromised child-rearing(e.g., leaving the child at home and going away);

- Maternal malnutrition during pregnancy, delivery, and postnatal period, and

- Maternal stress and being undermined by the husband (since this affects child-rearing).

*If the mother leaves the infant at home and goes away, if the mother has some problems if she doesn't eat well and isn't healthy during pregnancy, she will give birth to an unhealthy baby. The baby will grow unhealthy if the mother doesn't get appropriate care /services starting from pregnancy.* (Deder Father)

**Healthcare workers' explanations.** HCWs agreed that lack of appropriate breastfeeding was a key and also noted infant illnesses as immediate causes of malnutrition. They also agreed that maternal factors, including health, play roles:

*As far as a health issue is concerned, diarrhoea could cause malnutrition in infants. Besides, severe pneumonia could lead to malnutrition in infants' u6m. As well, if the mother has HIV/ AIDS or TB, this may expose the infant to malnutrition. So, mother's health problem could cause malnutrition in infants' u6m.* (Deder health center nurse)

HCWs stressed that poverty / poor socioeconomic circumstances contribute to the problem, adding that many of the mothers of malnourished infants u6m are malnourished themselves. They also agreed that family conflict coupled with low socioeconomic status results in the caregivers not having a good mentality to prepare and consume nutritious foods for themselves. The circumstance more prevails in Deder.

*Obviosuly, the socio-economic status/conditions of mother could be a factor for child malnutrition, as I said earlier.* (DederHEW)

The same HCW noted socioeconomic problems due to not being able to provide formula milk were in the rare cases that were needed:

*Under the difficult condition where the mother could not feed her breast milk, replacement feeding should be provided based on the standard. So, if the child is not offered formula feeding based on the standard that could also be a reason for the malnourishment of the infant.* (Deder HEW)

Some HCWs also noted traditional narratives about malnutrition still being prevalent:

*Parents may think that it is an evil eye; they might think it is "bu'aa" so may take them for traditional abdominal massage and go to traditional healers.*(Jimma under-five children's nurse)

## Theme 5: Detection and treatment of malnutrition in infants u6m identification

**Carers and community perspectives.** Physical and clinical signs and symptoms mainly identified malnutrition in infants u6m. These include thinness, weight loss, body swelling

(oedema), diarrhoea, vomiting, hair color changes, and skin rash. Many signs/symptoms were not specific to malnutrition but were general manifestations of health problems. Some respondents did not know what was specific to malnutrition; others correctly recognised that overlap with other conditions is expected.

*Malnutrition in u6m infants is manifested, firstly, when the physical condition of the infant deteriorates from time to time, we can suspect for malnutrition. Secondly, if the infant persistently cries, it may indicate malnutrition. As well, if the weight of the infant is not increasing overtime, it may show malnutrition.* (Deder mother)

**Healthcare workers' perspectives.** Unlike caregivers, HCWs had a detailed understanding of malnutrition. Some, however, reported difficulties in identifying malnutrition in infants u6m. Others from high-level health institutions explained that there are predefined standards, but acknowledged that these do not seem to be universally known:

*Though we are aware of on the occurrence of malnutrition, it is challenging to assess formal nutrition among u6m infants; this is because of a lack of set standards to screen/diagnose.* (Deder HEW)

*Obviously, there is a scientific standard that we use to identify a malnourished infant. We are using those standards in this hospital. A national guideline is adopted from the WHO to identify a malnourished infant. Some infants become very thin when you look at them. So, this could be one way even though it has some subjectivity. Secondly, we use the body mass index by taking the weight and length of the infant. So, there is a standard to check whether the infant is in the range of normal growth or malnourished area. There might be other methods also.* (Jimma paediatrician)

The commonest physical signs reported were hair colour changes, delayed development, body swelling, skin ulcerations, and wasted appearance. HCW also assesses the maternal illness and nutritional status, infant feeding pattern, early initiation of complementary feeding, and associated history of infant illness such as infant illness such as the history of diarrhoea, vomiting, and previous history of malnutrition.

*We use criteria, weight for age less than -2 SD (Z-score) using the current WHO criteria or presence of bilateral leg oedema and other pathologies. But before the full-blown features of malnutrition, we can ask the mother about nutritional history, the pattern of the feeding, initiation of complementary feeding, and associated illnesses. We can assess the patient as a whole and look for the risk factors and specifically for the signs of malnutrition.* (Jimma paediatrician)

HCW also reported that MUAC is currently not recommended by national guidelines for infants u6m, despite it being widely used for older children. After further research, they did see a potential role for MUAC, especially for community use.

*Firstly, we use weight measurement to identify malnutrition in u6m infants. The child's weight should match his age. Like a growth monitoring chart, we can check the weight to age ratio foru6m every month when they come for vaccination. So, based on the weight-to-age ratio results, we can identify whether the infant is malnourished or not. Secondly, we cannot diagnose the presence or absence of malnutrition in u6m by physical observation alone. So, we*

*use a growth monitoring chart to identify a malnourished infant from a normal one.*(Deder HEW)

*I think it is obvious that MUAC is helpful specially for a community screening of child malnutrition. It is fast and easy to use. It is useful to identify malnutrition in the first stage (for categorization) of mass screening. But as I said, currently, it is not recommended for infants u6mbecauseof its limitation on the issue of sensitivity. (*Jimma, paediatrician)

Some HCW also saw identifying malnutrition in infants u6m as difficult and time-consuming:

*Actually, it is difficult to detect malnutrition in u6m. It also takes a bit longer as compared to infants older than 6 months. It is easier to identify malnutrition in children older than 6 months.* (Jimma HEW)

Monthly screenings for malnutrition were done at Health Posts (HPs) and/or Health Centres (HCs); small and nutritionally at-risk infants were referred to higher-level facilities.

*By just looking at them, you will target them for additional screening, or you can just say this one has malnutrition. . . However, most often, weight for age is used to identify malnutrition and we don't use MUAC.* (Jimma MCH Coordinator)

HCW also reported some challenges with weight and length measurements:

- Unavailability of infant scales,

- Shortage of staff,

- Time consuming nature of assessments, and

- Requirements for specific technical skills

Despite this, some preferred these measurements over MUAC as they believe that weight and length have better diagnostic ability than MUAC. Others preferred MUAC due to its simplicity, irrespective of diagnostic limitations.

*I think MUAC is easy and fast, but it doesn't properly diagnose the presence of malnutrition in infants.* (Jimma paediatrician)

*MUAC is very easy compared with taking weight and length and checking these on growth standards. Additionally, the latter needs weighing equipment and length board whereas MUAC doesn't. . .there is nothing the HEWs need to carry if they can use MUAC. (*Deder MCH officer)

### Treatment of malnutrition

**Preferences for seeking treatment.**    Both carers and HCWs report that most parents visit health centers if they suspect malnutrition in infants u6m. They also noted that proximity to care is valued, hence a preference for treatment at HP level–but only as long as the quality of the service is not compromised. Since HPs are lower-level facilities with fewer resources, some concerns were raised about quality of care (lack of technical skills and medical supplies). In contrast, higher-level treatments at health centers and hospitals were perceived as better quality but costly:

*It would have been very nice if the treatment services were provided by HEWs at a health post in their locality, but there is no such service at the local level. If people go to hospital or health center, they will incur unnecessary costs like transportation food, etc. Thus, the first choice of people to get health services is in their locality. (*Deder father)

*Health extension workers have no technical skills, capacity and medical supplies to treat malnourished infants at the health post. They can only refer them to the health center. (*Deder father)

Reportedly, seeking treatments with other healing (traditional) practices than health facilities and home treatment is rare.

**Treatment approaches as per the standard protocol.** Both the carers and the healthcare workers reported that communities usually take infants with health problems, including those with suspected malnutrition first to HEWs at the health posts (these are the first step of the healthcare pyramid). They also reported that for the u6m infants with malnutrition, HEWs only counsel and then refer them to the nearest health center (the next level up in the overall healthcare system). The healthcare workers reported that at some of the health centers, there is a dedicated inpatient area for malnourished infants where long side breastfeeding, therapeutic milk (F75/F100) will be given. They also mentioned that after successful treatment at the health centers, the infants are sent with feedback to HEWs so that they provide monitoring and follow-up care; hospital referrals are made in cases of severe illness, or if health center staff can't manage a complex case (e.g., if they are not confident with intravenous fluids) as per the healthcare workers reports.

**Treatment-related challenges.** The healthcare workers reported the HEWs' limited capacity to identify early-stage u6m infant malnutrition as a key challenge. Consequently, many carers go directly to health centres or hospitals by passing the HEWs and health posts. But since these health centers and hospitals are often being far away, some do not go at all, so they do not get any meaningful treatment.

In order to overcome these challenges, HCWs suggested that identification should start at the HP level by the HEWs and the treatment should be given at health centers and hospitals through referral linkages and they noted that. Currently, these linkages are not strong. They suggest that HEWs should be empowered to use all assessment methods in parallel to health centres and hospitals.

Another challenge reported by the health workers is the fact that not all health centers can treat malnourished infants which either refer malnourished infants u6m to local hospitals or just advise mothers to exclusively breastfeed; one of the reasons reported for this by a HCW was the unavailability of supplementary feeds like milk top-up at the health centers. One more challenge reported by the HCW is that mothers are reluctant to stay with their infants (when the infants u6m with malnutrition are admitted at the health centers or hospitals) due to work pressures:

*The main problem, especially in Deder, is mothers' inability to stay with their infants to breastfeed. They leave their infants at a very early age and return to work. There are even mothers who go to the khat market leaving their malnourished infants at the health centre not to miss their khat business. As a result, mothers complain about the inpatient treatment of infants with malnutrition. Even at the hospital, mothers don't want to stay for a couple of days looking after their babies . . .. Sometimes they escape from the hospital by quitting treatment of their infants before the discharge paper is issued to them by the concerned physician.* (Deder health center nurse)

Both the carers and the HCWs in both study sites reported that there are some mothers who decline hospital referral due to other financial constraints. Additionally, they reported that sometimes referral delays occur because parents need to raise money for treatment-related costs.

Lastly, HCWs mentioned that they are worried about the negative influence of using the bottles to give diluted F100 in hospitals:

> . . ..health workers at hospitals provide the diluted F100 with a bottle and the community sees that. When we and the HEWs tell the mothers in the community not to use bottle feeding, they refuse because they think the knowledge and practices at the hospitals are better and say the HEWs in the rural area are less knowledgeable. (Deder MCH Coordinator)

## Discussion

This qualitative study explored important issues regarding communities' and HCWs' perceptions around small and nutritionally at-risk (malnourished) infants u6m. Five key themes emerged from our interviews: understanding of normal infant health and wellbeing; perceptions about infant feeding; awareness about malnutrition; understanding of reasons for malnutrition; how to identify and treat affected infants and their carers. Under each theme, we identified both barriers and facilitators that might hinder or help future treatment programmes under each theme. These results will help work on infant malnutrition in Ethiopia in the short term. More importantly and in the medium to long term, our theme list is widely generalizable. It forms a framework that others working on the same topic in other countries and other settings can use to profile and better understand their contexts. Our findings are thus relevant to a broad audience, including policymakers, program implementers and researchers, helping them design and implement programs and research for a vulnerable and needy group of infants. More specifically, this finding is a path-opening step to effectively implement the MAMI trial project in the local and contextual bases. It explored the detail of community's, healthcare workers' and program managers' overall understanding and perspectives in managing malnourished u6m infants.

Our community defined an "ideal infant" based on physical appearance and behaviours. Being "fat", alert, active, burping, and crying (but not excessively), are considered good health and nutrition signs. On the plus side, this demonstrates good understanding of the signs and symptoms of good health and hence, carers will do the necessary things like burping which will be beneficial for the infant and also the refrain from doing some unnecessary interventions like seeking for treatment or introducing complementary feedings for some of the normal behaviours like crying (often people assume infant's cry due to illnesses and when the breast milk is not enough). Understanding the range of age-appropriate sleep patterns for an infant can reassure a caretaker that their infant is healthy and well and can also help identify those infants with an underlying problem that require further evaluation. This will ultimately avoids/minimizes myths/misconceptions around sleep and hence unnecessary interventions or omission of necessary care and support when needed.

However, a potential associated problem is that the signs are not specific; over-reliance on these physical signs and symptoms might result in vulnerable infants being missed and not getting the healthcare/nutrition services they need; for example infants crying excessively might be having pain due to different kinds of illnesses like middle ear infections which might be overlooked if both the carers and HCWs are not vigilant enough in appropriately evaluating the infant. An important study from Kenya found that verbal descriptions of under nutrition were inferior to pictorial scales and strongly recommended that objective anthropometric

tools be used to assess nutritional status [17]. A review also found that clinical assessment alone is a poor way of screening for and identifying malnutrition [18]. The preference for a "fat" infant might itself be problematic since there might be a temptation to add supplementary milk or other feeds to an infant who is small, lean but otherwise healthy and growing well– both carers and healthcare workers are susceptible to this preference for body size over healthy growth [19]. Beyond physical signs, symptoms and anthropometric indices, the healthcare workers rely on the maternal history during and after pregnancy as additional criteria to define an "ideal infant." Previous studies also reported that maternal health-related issues during and after pregnancy such as her decision making role on her health, antenatal care follow-up along with the place of delivery, maternal malnutrition [20],maternal mental health and well-being are essential characteristics in determining the "ideal infant" and vice versa [21, 22].

Our study participants had good knowledge of infant u6m feeding practices. However, there were some exceptions, with tensions noted between what people know and what they do. Most participants explained that everybody knows that exclusive breastfeeding up to six months has an indispensable role in having "a well, physically growing and healthy infant." However, they also reported several of common circumstances where there was early introduction of other foods, mainly water, and other semi-solid foods. This included mothers going out to work emphasizes the need for support programs to consider broader family circumstances that might affect infant nutrition/health. One notable local practice centered around a belief that at infant's fluid needs cannot be met by breast milk alone and that water top-ups were needed. This was strongly and commonly held and is a significant barrier that needs to be addressed in any breastfeeding-related work in Ethiopia. Even if mothers understand that infants require no additional water during exclusive breastfeeding, they need support to resist family pressure on adverse practices. This may be a reason why breastfeeding programes involving fathers are more likely to succeed [23]. Our findings of a variety of factors influencing feeding are similar to other work which reported that: some mothers have poor knowledge on infant feeding practices and perceive that breast milk alone is inadequate [24]; those with strong religious narratives [25–27] often start early complementary foods for infants u6m. Even if the good knowledge reported by the study participants can be taken as an opportunity, significant efforts have to be put in place to counsel and support the mothers of infantsu6mto-putthisknowledgeintopracticeso that the infants u6m could be on exclusive breastfeeding. Additional major challenges identified not to translate the good knowledge and attitude regarding exclusive breast feeding in our study was the lack of adequate support (health, nutrition and economic) for the mother. So, for programs to be successful in preventing malnutrition in infants u6m, interventions targeting to address the health and wellbeing of the mother including reducing the stress on the mother, routine evaluation of the mother for mental and other medical illnesses and providing some economic supports as well as extended maternity leave for the employed mothers should be considered. Some of these interventions might be beyond the capacity of the health sector and hence need engagement with other relevant stakeholders.

This again was good regarding awareness of malnutrition among both community members and healthcare workers. Healthcare workers in particular were able to offer detailed explanations. It is encouraging and likely reflective of good health-related communication that those mothers who connected with healthcare services came away with improved understanding. On the barrier side, local languages were reported as having limited terminology to express malnutrition-related issues with signs and symptoms of malnutrition often mixed up with other signs and symptoms of poor health. This highlights the importance of continued health education, ideally using various tools (e.g., posters and other visuals) to increase community understanding and improve related practices [28, 29]. Knowledge of the reasons

underlying and associated with malnutrition was overall positive and is an essential facilitator for future programs to build on. It was good that most of our respondents recognised the importance of breastfeeding towards good nutritional status and avoiding malnutrition. However, the wider numbers of other possible causes were not well recognised. With the correct identification of breastfeeding problems as a cause of malnutrition comes an associated risk that carers and healthcare workers alike will forget that many other underlying causes are also possible. Beyond breastfeeding problems as cause of malnutrition, this study also identified a strong belief in the community that consumption of fruits and animal food sources during pregnancy leads to excessive fetal growth which predisposes for caesarean section delivery. However, previous studies reported that consuming such food groups crucial in reducing the risk of small- for-gestation babies and improving foetal growth during pregnancy [30, 31]. This is why current and future programs—including our own MAMI care pathway [30]–must actively consider and look for and support other issues alongside breastfeeding. There is still unfinished agenda and need for improved nutrition education related to infants u6m. Traditional beliefs such as the evil eye might be hard to deal within the short term: this must be done with tact and sensitivity not to alienate those who firmly hold to these [31]. Others, like poor socioeconomic circumstances being a significant risk factor for infant malnutrition are also hard to deal with by the healthcare workers alone because they require broader societal change and several stakeholders' engagement. This highlights the need for more comprehensive advocacy efforts alongside patient-focused clinical health and nutrition work.

Better future identification and treatment of malnutrition among infants u6m should capitalize on the fact that community members and healthcare workers are aware of some of the key signs and symptoms. This includes awareness of some key anthropometric measures. Though indices like weight-for-length, recommended in current WHO severe malnutrition criteria [11], are already well recognised, limitations are important. Given the time, expertise, and equipment needed weight-for-length is not often measured And when it is, measurement quality is poor: there is an urgent need to consider other measures, including MUAC(Mid upper arm circumference).This is quick, easy, reliable and can readily be done in the community [32, 33]. There is growing evidence that MUAC and weight-for-age are better than weight-for-length to identify infants u6m at high risk of mortality [18, 34–40].

An important challenge to the success of treatment programs for infant u6m malnutrition is that carers often prefer health centers, hospitals and other higher-level facilities than taking them to traditional practitioners. This is due to a perception that better treatments are available there compared to outpatient care; similar findings were reported from Niger [41, 42] and Bangladesh [16].Paradoxically, however, they rarely attend when referred–factors like distance and costs play a part in this [42]. Ideally, community and higher- level facilities should be well resourced and offer effective treatment, with referral and back-referral between them according to the clinical need. However, in many resource-constrained settings, this does not work well due to the challenges associated with the skills of care providers, availability of the necessary medical supplies, and lack of set standards to manage malnutrition in infants u6m, transportation and food- related costs.

A strength of our study is that in exploring issues from one setting in- depth, we have developed a conceptual framework and template, which we hope, can be helpful for others working on malnutrition in small and nutritionally at-risk infants u6m. Though each setting will have some unique barriers and facilitators, many similarities are also likely. In particular, key stakeholders whose views and preferences must be understood will always include carers and community members, local healthcare workers, managers and programmers running health and nutrition services. As we have done here, consulting them has a key role in designing better future research programs and nutrition/health programs for infants u6m. Using a similar

approach, a similar topic guide, and interviewing similar key informants, others can understand their context and hence tailor their infant's nutrition programs accordingly.

Another strength of our study is including two very different contexts and geographic areas of Ethiopia. Most of the findings were similar despite their differences, making further generalizability also likely, both in Ethiopia and beyond. Finally, we included a good range and number of participants exploring different perspectives and practices.

We also acknowledge the limitations of our study. While we identified many relevant beliefs and practices related to infant u6m malnutrition, we do not know how widespread they are. This would need future work and different (quantitative) methodologies to determine. We also cannot be sure that community members shared all topics: despite our interviewers being Ethiopian themselves, they are all highly educated professionals. Given that we found a gap between what people say they know and what they do (e.g., knowing about but often not following national/international recommendations on exclusive breastfeeding till age six months), it is not impossible that there were other more sensitive topics or local beliefs that they were reluctant to discuss at all. Perceived or actual judgments placed on mothers by themselves or by some health workers when they do not follow 'best practice' may affect the relationship between caregivers and their healthcare workers and our interview responses.

Hence, future programs must ensure a trusted and non-judgmental relationship between mother and health worker is essential to allow for transparency and openness on 'real' versus 'reported' practice. These matters regarding the accuracy of feeding information and to ensure a mother gets the support she needs to address challenges. Even though we tried to minimize reporting bias by elaborating the study's aim and purpose, some participants might have provided socially desirable responses. Finally, we initially planned to conduct focus group discussions but did not do so because of the COVID-19 related risks of gathering groups together. It is possible–albeit unlikely–that other themes may have emerged.

Future research related to this study includes:

- They repeat similar qualitative work in other settings, both Ethiopia and other countries, to document similarities and differences. Future work will be made easier and will be of more use to policymakers, program managers and researchers if those future papers follow a similar analytical /topic framework as we have done in this work.

- Quantitative surveys to assess the prevalence of barriers and facilitators we have identified.

- Developing and testing communication and behavior-change materials to address specific barriers(e.g., tackling the wrong idea that infants u6m need water in addition to breastmilk)

- Developing and testing interventions broader care packages to manage small and nutritionally at-risk (malnourished) infants u6m. (We are planning such a randomised controlled trial in Ethiopia towards the end of 2021, but others must do likewise in other settings).

## Conclusions and recommendations

Despite good carer, community and healthcare worker knowledge of some factors underlying infant u6m malnutrition (notably regarding breastfeeding), our study has highlighted several issues, which must be addressed in order to tackle the problem. Some might be specific to our setting (e.g., the solid cultural imperative to give water in addition to breastfeeding) while others might be more generalizable (e.g., interrupting breastfeeding before six months due to maternal work commitments; mismatch between health/nutrition knowledge and actual practices; conflict between traditional and scientific narratives; a perception that hospital-based care is superior to community-based care).

To optimise the chances of positive impact, research and programs managing small and nutritionally at-risk infants u6m must understand and make the best of the setting they work in; building on community/healthcare system strengths; address common myths/misconceptions and problems. Additionally, beyond working on knowledge and attitude of the carers and HCW, different practical and context specific initiatives should be taken to support mothers having infants under 6 months so that they can exclusively breast feed their infants u6m and prevent occurrence of malnutrition and associated complications. Some of these initiatives include supporting the health and wellbeing of the mother including maternal mental health, providing economic support for the in-need mothers and also providing extended maternity leave for the employed mothers by involving the different stakeholders outside the health sector.

## Supporting information

**S1 Text. In-depth interview guide for carers.**
(DOCX)

**S2 Text. Key informant interview guide for health workers.**
(DOCX)

**S1 File.**
(DOCX)

**S2 File.**
(DOCX)

## Acknowledgments

The authors are thankful to the study participants for their kind participation. We are grateful to Abdulhakim Ali for field coordination during data collection in Deder. We also thank all the personnel at the zonal and district health offices and leaders of the health facilities for facilitating their support during the data collection.

## Author Contributions

**Conceptualization:** Marie McGrath, Marko Kerac.

**Data curation:** Nega Jibat, Mubarek Abera, Melkamu Berhane.

**Formal analysis:** Nega Jibat, Ritu Rana, Ayenew Negesse.

**Funding acquisition:** Marie McGrath, Marko Kerac.

**Investigation:** Nega Jibat, Ritu Rana, Ayenew Negesse, Mubarek Abera, Alemseged Abdissa, Tsinuel Girma, Anley Haile, Hatty Barthorp, Marie McGrath, Carlos S. Grijalva-Eternod, Marko Kerac, Melkamu Berhane.

**Methodology:** Nega Jibat, Ritu Rana, Ayenew Negesse, Mubarek Abera, Alemseged Abdissa, Tsinuel Girma, Anley Haile, Hatty Barthorp, Marie McGrath, Carlos S. Grijalva-Eternod, Marko Kerac, Melkamu Berhane.

**Project administration:** Marko Kerac, Melkamu Berhane.

**Resources:** Ritu Rana, Ayenew Negesse, Mubarek Abera, Alemseged Abdissa, Tsinuel Girma, Hatty Barthorp, Marie McGrath, Carlos S. Grijalva-Eternod, Marko Kerac, Melkamu Berhane.

**Software:** Ayenew Negesse.

**Writing – original draft:** Nega Jibat, Ritu Rana, Ayenew Negesse.

**Writing – review & editing:** Nega Jibat, Ritu Rana, Ayenew Negesse, Mubarek Abera, Alemseged Abdissa, Tsinuel Girma, Anley Haile, Hatty Barthorp, Marie McGrath, Carlos S. Grijalva-Eternod, Marko Kerac, Melkamu Berhane.

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
