## [Decision Letter · Decision Letter 0]

9 Feb 2022

PONE-D-21-13630Carers’ and health workers’ perspectives on malnutrition in infants aged under six months in rural Ethiopia: A qualitative studyPLOS ONE

Dear Dr. %Jibat%,

Thank you for submitting your manuscript to PLOS ONE. After careful consideration, we feel that it has merit but does not fully meet PLOS ONE’s publication criteria as it currently stands. Therefore, we invite you to submit a revised version of the manuscript that addresses the points raised during the review process.

Please respond to each comment made by each reviewer and a thorough proof reading is performed.

We look forward to receiving your revised manuscript.

Kind regards,

Mary Hamer Hodges, MBBS MRCP DSc

Academic Editor

PLOS ONE

Journal Requirements:

2. Please include a copy of the interview guide used in the study, in both the original language and English, as Supporting Information, or include a citation if it has been published previously.

5. Your abstract cannot contain citations. Please only include citations in the body text of the manuscript, and ensure that they remain in ascending numerical order on first mention.

Additional Editor Comments (if provided):

Please deal with each comment form each reviewer before resubmitting.

Reviewers' comments:

Reviewer's Responses to Questions

**Comments to the Author**

1. Is the manuscript technically sound, and do the data support the conclusions?

Reviewer #1: Yes

Reviewer #2: Yes

Reviewer #3: Partly

2. Has the statistical analysis been performed appropriately and rigorously? 

Reviewer #1: N/A

Reviewer #2: N/A

Reviewer #3: N/A

3. Have the authors made all data underlying the findings in their manuscript fully available?

Reviewer #1: Yes

Reviewer #2: Yes

Reviewer #3: No

4. Is the manuscript presented in an intelligible fashion and written in standard English?

Reviewer #1: Yes

Reviewer #2: Yes

Reviewer #3: Yes

5. Review Comments to the Author

Reviewer #1: General comments: Good study to assess understanding in the community perspective especially in preparation for a trial. The results section can be shortened by combining HW and Community members perspectives. As it is now there is a lot of repetition that can be avoided. The authors can consider presenting the common responses from the two groups whilst still highlighting differences in opinion between the two groups. The results section is currently 18 pages and difficult to get through. This can be shortened to improve on its readership.

In the methods section, I take note of the fact that the data was collected by 3 people. The authors should consider clarifying why this was necessary especially because it implicates reliability and validity of the data. Clarify how did you decide on who interviews who?

In the data collection section authors can clarify and add details on how interview tools were developed, translated and back translated, was it piloted before application?

Authors should also describe the COVID related considerations that were applied during data collection in 2020?

In the discussion, authors can comment about how the results/findings may be used to support implementation of the cRCT mentioned in the introduction.

Reviewer #2: Overall, this is an interesting manuscript that presents the results of a qualitative study evaluating perceptions of malnutrition among infants less than 6 month of age at two sites in Ethiopia. The results are interesting and may be of interest to those working in nutrition programs as well as primary care settings in LMIC settings.

The abstract frames this work in a way that seems very narrow and limiting - understanding these challenges is important for many reasons - not just to inform research - I think the abstract sells the paper short and makes it feel far less relevant than it could. The idea that interventions only need to address misunderstandings and barriers to understanding is also problematic. I would strongly encourage the authors to take a broader view of this work and its implication for research, practice, policy

In the introduction, the authors suggest that supporting BF is the core of treatment. In may be worth noting that evidence from several studies (Mwagombe et al, etc) suggests that BF alone is not sufficient for recovery in severely malnourished infants.

In the last paragraph of the introduction - I am not sure I would structure this paper around the needs of the trial - that may be true, and worth mentioning, but as noted above the implications of this work are much broader

Overall, more detail on how participants were selected (and a discussion of the inherent limitations to this approach) is needed.

What is the difference between the face-to-face vs. in-depth interview?

In the results section for Theme 1, there are a lot of behaviors being mentioned that are quite age dependent. A one month old has very different feeding, playing, eating behaviors than a 6 month old. Was there a sense that the participants recognized that these behaviors differed by age?

On page 10, “Caregivers also consider that infants should sleep in a clean place where there are no insects.”. This seems more like a feature of the environment, not the child.

Again, the lack of age appropriate behavior in the discussion is striking - was this evaluated? It seems very hard to interpret behaviors that differ by age.

Page 13, “there is a community belief that eating animal products and fruits will lead to excessive foetal growth and hence predispose to caesarean section delivery.” This is a well established phenomena - lots of data from Nepal about this - would be worth discussing more in detail.

Throughout the results sections (particularly in Theme 3), the authors present data as a synthesis of findings and not really results, please present objective findings in results and reserve the synthesis of findings for the discussion.

On page 18, the authors note that there is confusing between nutritional status and feeding as well as confusion about BF. These are confusing to many people and it is not clear what this has to do with education. In addition, please move the discussion of why this may be to the discussion and present objective results only in results.

For many of the HCW interviews it appears that there were some misconceptions. Important to note where the HCWs actually had erroneous impressions of what the guidelines say.

Page 22 – top of the Theme 5 section is repetitive with above section.

Page 24 – please define abbreviations (HP – Health Post) when first using.

Bottom of page 25 - This tendency to blame mothers for malnutrition is a major barrier and should be addressed and discussed. This leads to a breakdown of the critical alliance between HCWs and mothers and is a major issue.

Overall, it appears from these findings that education and knowledge are not the problem - mothers and HCWs knew what they were supposed to do, but didn’t do it. What other interventions are needed beyond KAP interventions?

Reviewer #3: Thank you for the opportunity to review this manuscript which reports on a qualitative study “assess the perceptions and understanding of malnutrition in infants u6m and its management among carers, communities and healthcare workers in rural Ethiopia”. The study was conducted as formative research to inform a cluster randomized trial on management of malnutrition among young infants.

Introduction

“An estimated 8.5 million infants u6m worldwide are wasted (have low weight-for-length, a marker of severe malnutrition)” please clarify if you mean “An estimated 8.5 million infants u6m worldwide have weight-for-length Z scores <-3, a marker of severe malnutrition.” Ie is the 8.5 million among those with “moderate or severe wasting” or severe wasting only. Given the focus of the study, it would be best to present the data restricted to severe wasting.

I believe the word conducive is intended rather than conductive in this phrase “requiring skilled professionals and a conductive environment”

In the list of practices and how prevalent they are on the top of page 5 “9% were given a bottle with a nipple” is listed in the midst of other practices related to TYPES of feeds – this items seems misplaced here.

Methods

Domain 1 is well explained. My only question is related to the choice of the word ”authors” under personal characteristics. Did the entire research team co-author the manuscript or does the word choice need revision?

Domain 2 requires more context. For example either in the Methods or the Intro please describe health developmental army. HEWs, health managers, and program managers within the Ethiopian health system context. The readership will include persons outside of Ethiopia. For ex, are health managers the administrative heads of a district health management team? What is their positionality to program officers, etc.

What is meant by carers? Are these caretakers of U6m? How were they and HDA members identified as interview candidates? I note the purposive sampling but we still need to understand how much bias is involved in the selection process. I’d like to know more about the eligibility criteria for all of the key informant types. Eg did you seek HCWs who manage sick and small infants? Were all parents approached because they had a child in a hospital?

What is the difference between key-informant and in-depth interviews? Ie why the “or” in “key-informant interview or in-depth interview”

Domain 3 is well explained however, a statement as to whether a single coder per transcript was use or dual coding in which case how differences in coding between coders was resolved.

Please submit a Consolidated criteria for reporting qualitative studies (COREQ) checklist as supplementary material. This will help the reader understand if this work was conducted according to standards for qualitative research.

RESULTS

Recommend merging the 1st 2 sentences: “We conducted a total of 31 interviews: 17 with mothers, fathers, and HDA leaders, and 14 with healthcare workers and managers (Table 1).” What about the program officers and HEWs that were mentioned in the methods? Are the mothers and fathers from the same household? How many health facilities are the HCWs from? ~What % of HCWs in the Jimma and Deder are represented by the interviewees?

Are the descriptions of the 2 people who refused examples of refusals or the only 2? It’s a bit of an unusual way to report refusal rate.

Table 1 – In the methods section you note that the following types of health care workers were targeted for interviews: nurses, midwives, health officers, paediatricians, and HEWs. But Table 1 only includes the latter 2 which represent very different cadres. What about the nurses, midwives and HOs?

Table 1 – were all of the fathers also HDA members? They are co-listed

Can you provide some basic demographic data about these interviewees? Eg ages, sex of HCWs

Theme 1

It is difficult to understand how these very general interview responses on general health and development help us understand perceptions around management of severe malnutrition of U6m. The results are so general and sweeping but its hard to image how these will inform interventions.

Theme 2

The data in the first section (p 13 and thru p 14 before the Reasons for early initiation of additional feedings section are hardly novel or surprising. The Reasons for early initiation of additional feedings section is more specific and adds to the literature and can help inform future work. The first bullet point re mother’s health and economic conditions deserves more elaboration.

Theme 3 is an important section. But without more context on characteristics of the parents interviewed were are not able to fully appreciate these results. For ex, it is stated “Parents who actually had small or malnourished infants had a better understanding of related concepts because of their contact with health facilities during treatment.” But what % of parents interviewed had malnourished infants? See comments above re this issue.

Treatment approaches as per the standard protocol – p 24 – is this the authors’ explanation of guidelines in Ethiopia or a report from the interviews? Is F75/F100 really recommended for management in <6 month olds and in health centers as opposed to in hospitals? Is there a reference for this guideline (if this is what it is)?

Treatment related challenges

“HPs’ limited capacity to identify early-stage infant u6m malnutrition was reported as a key challenge.” – reported by which type of interviewees?

2nd paragraph – again, I’m not sure if this information about norms is supposed to be background provided by the authors or if they are reporting data from their interviews.

Discussion

Through the bottom of page 29 most of the text is a rehash of the results with little suggestion of how these data are helpful in informing their future RCT, for example. Top of page 30 is a description of other literature. My point is that the discussion does not synthesize the results in a way that helps the reader understand how these results can be used, how they can inform next steps and additional work. The aim of the study was noted to inform a cluster randomized trial on management of malnutrition among young infants but it's not clear how - this requires some explanation.

Bottom of p 29 – “carers often prefer health centres, hospitals and other higher-level facilities due to a perception that better treatments are available there” – compared to what?? Community based care?

A thorough proofreading of the manuscript is necessary to catch all grammatical errors, some examples: in the Ethics section: “We obtained written consent from all study participants with language of their interest”. Another example from the Introduction where there is a tense mismatch within the sentence: “Whilst exclusive breastfeeding was practiced by 59% of mothers overall, declining sharply with age:”. Another example from the results section: “A common community belief says that infants cry for food looking when others feed”

6. PLOS authors have the option to publish the peer review history of their article (what does this mean?). If published, this will include your full peer review and any attached files.

Reviewer #1: **Yes: **Dr Martha Mwangome

Reviewer #2: No

Reviewer #3: No

---

## [Author Response · Author response to Decision Letter 0]

13 Apr 2022

Most comments of the reviewers' are accepted and incorporated. Clarifications are given where required. Details of the responses are indicated in the revised manuscript with track changes and in the rebuttal letter. We are thankful for the reviewers for their critical and constructive comments.

---

## [Decision Letter · Decision Letter 1]

7 Jul 2022

Carers’ and health workers’ perspectives on malnutrition in infants aged under six months in rural Ethiopia: A qualitative study

PONE-D-21-13630R1

Dear Dr. %Jibat%,

We’re pleased to inform you that your manuscript has been judged scientifically suitable for publication and will be formally accepted for publication once it meets all outstanding technical requirements.

Kind regards,

Mary Hamer Hodges, MBBS MRCP DSc

Academic Editor

PLOS ONE

Additional Editor Comments (optional):

Thank you, A nice job in responding to reviewers comments.

Reviewers' comments:

Reviewer's Responses to Questions

**Comments to the Author**

1. If the authors have adequately addressed your comments raised in a previous round of review and you feel that this manuscript is now acceptable for publication, you may indicate that here to bypass the “Comments to the Author” section, enter your conflict of interest statement in the “Confidential to Editor” section, and submit your "Accept" recommendation.

Reviewer #2: All comments have been addressed

2. Is the manuscript technically sound, and do the data support the conclusions?

Reviewer #2: Yes

3. Has the statistical analysis been performed appropriately and rigorously? 

Reviewer #2: N/A

4. Have the authors made all data underlying the findings in their manuscript fully available?

Reviewer #2: Yes

5. Is the manuscript presented in an intelligible fashion and written in standard English?

Reviewer #2: Yes

6. Review Comments to the Author

Reviewer #2: The authors have done a nice job responding to the initial reviewer comments. I am pleased to see that careful attention was paid to the comments.

7. PLOS authors have the option to publish the peer review history of their article (what does this mean?). If published, this will include your full peer review and any attached files.

Reviewer #2: No

---

## [Editor Report · Acceptance letter]

12 Jul 2022

PONE-D-21-13630R1 

*Carers’ and health workers’ perspectives on malnutrition in infants aged under six months in rural Ethiopia: A qualitative study*

Dear Dr. Jibat:

I'm pleased to inform you that your manuscript has been deemed suitable for publication in PLOS ONE. Congratulations! Your manuscript is now with our production department. 

Kind regards, 

on behalf of

Prof. Mary Hamer Hodges 

Academic Editor

PLOS ONE